# Correlation between Adrenoceptor Expression and Clinical Parameters in Degenerated Lumbar Intervertebral Discs

**DOI:** 10.3390/ijms232315358

**Published:** 2022-12-05

**Authors:** Marco Brenneis, Zsuzsa Jenei-Lanzl, Johannes Kupka, Sebastian Braun, Marius Junker, Frank Zaucke, Marcus Rickert, Andrea Meurer

**Affiliations:** 1Department of Orthopedics (Friedrichsheim), University Hospital Frankfurt, Goethe University, 60590 Frankfurt, Germany; 2Dr. Rolf M. Schwiete Research Unit for Osteoarthritis, Department of Orthopedics (Friedrichsheim), University Hospital Frankfurt, Goethe University, 60590 Frankfurt, Germany

**Keywords:** intervertebral disc degeneration, sympathetic nervous system, adrenoceptor expression, spine, Pfirrmann classification, Modic classification

## Abstract

Despite advanced knowledge of the cellular and biomechanical processes of intervertebral disc degeneration (IVDD), the trigger and underlying mechanisms remain unclear. Since the sympathetic nervous system (SNS) has been shown to exhibit catabolic effects in osteoarthritis pathogenesis, it is attractive to speculate that it also influences IVDD. Therefore, we explored the adrenoceptor (AR) expression profile in human IVDs and correlated it with clinical parameters of patients. IVD samples were collected from n = 43 patients undergoing lumbar spinal fusion surgery. AR gene expression was analyzed by semi-quantitative polymerase chain reaction. Clinical parameters as well as radiological Pfirrmann and Modic classification were collected and correlated with AR expression levels. In total human IVD homogenates α1A-, α1B-, α2A-, α2B-, α2C-, β1- and β2-AR genes were expressed. Expression of α1A- (r = 0.439), α2A- (r = 0.346) and β2-AR (r = 0.409) showed a positive and significant correlation with Pfirrmann grade. α1A-AR expression was significantly decreased in IVD tissue of patients with adjacent segment disease (*p* = 0.041). The results of this study indicate that a relationship between IVDD and AR expression exists. Thus, the SNS and its neurotransmitters might play a role in IVDD pathogenesis. The knowledge of differential AR expression in different etiologies could contribute to the development of new therapeutic approaches for IVDD.

## 1. Introduction

Low back pain (LBP) can lead to severe physical and mental impairment in patients’ quality of life. More than 600 million people worldwide suffer from LBP and globally about 70–85% experience LBP at some time in their lives [1]. Despite intensive research, etiology and the factors that trigger an episode of LBP remain unclear. This limits the possibility of designing effective preventive and therapeutic strategies [2]. Despite inconsistent research results, lumbar intervertebral disc degeneration (IVDD) was found to be a major risk factor for LBP [3,4,5,6]. Similar to LBP, the etiology of IVDD is not fully understood and described as “multi-factorial”. Thus, biomechanical wear and tear, disturbance of physiological cellular behavior and genetic inheritance are interdependent causes of degenerative processes in the intervertebral discs (IVD) [7,8,9,10,11]. In spite of increasing knowledge of cellular and biomechanical processes, the exact trigger as well as mechanisms of IVDD and phenomena such as early adjacent segment degeneration (ASD) remain unclear. Previous studies have shown that the function of the IVD highly depends on the interaction between mechanical behavior and cellular responses [12,13,14].

In this context, analyzing crucial components of the sympathetic nervous system (SNS) could provide a better insight into the pathophysiology of IVDD. Previous studies elucidated that all joint tissue expresses adrenergic receptors (AR) and that the SNS in fact is involved in bone and cartilage metabolism and remodeling [15,16]. Besides the ingrowth of nerve fibers into articular cartilage during the progression of osteoarthritis, a catabolic effect of norepinephrine (NE) via α- and β-AR on cellular homeostasis in murine joint cartilage has been demonstrated [15,17,18,19]. With regard to the fact that physiology and extracellular matrix in degenerated articular cartilage and IVD tissue share several similarities [20], Barczewska et al. proved the presence of sympathetic nerve fibers in degenerated bovine IVD-tissue [21]. On the basis of these results, Kupka et al. were able to detect AR gene expression in tissue of degenerated human IVDs and an induction of intracellular signaling pathways through the sympathetic neurotransmitter NE that activates ARs [22].

The aim of the present study was to correlate the expression level of the detected ARs in IVDs with clinical parameters of the patients such as radiological degree of degeneration, preoperative pain level and medication. An increased knowledge of the relationship between those clinical parameters and the AR expression and AR-dependent signaling pathways in IVD tissue could contribute to the development of novel therapeutic strategies for IVDD treatment.

## 2. Results

### 2.1. Patient Characteristics

Patient characteristics (n = 43) were examined retrospectively. The mean age of the patients was 69.23 (SD:7.66) years and the mean Body Mass Index (BMI) was 30.77 (SD:5.67) kg/m^2^. According to the Numeric Pain Rating Scale (NPRS) patients had a mean preoperative pain of 6.81 (SD:1.53). Altogether, seven patients (16.3%) had a cranial or caudal ASD after previous spinal fusion surgery. The largest proportion of the IVD samples was collected from the segment L4/5 (34.9%). No significant difference in α1A-, α2A- and β2-AR expression was found between the respective segments. Table 1 shows the characteristics of the patient cohort.

### 2.2. AR Gene Expression and Modic Classification

Patients were divided into two groups [no structural degeneration of the endplates (Modic 0-I) vs. structural degeneration of the endplates (Modic II-III)] to investigate whether structural alterations of the endplates correlate with the relative AR expression of the respective IVD. Results are shown in Figure 1. Significant differences in α1A- (*p* = 0.009; R = 0.3997), α2A- (*p* = 0.043; R = 0.3200) and β2- (*p* = 0.050; R = 0.3003) AR expression were found when analyzing IVDs with intact endplates (Modic 0-I; n = 19) in comparison with structurally degenerated endplates (Modic II-III; n = 23). The median of differences in relative α1A-AR expression between both groups was −0.35 (CI: −0.71 to −0.07), in relative α2A-AR expression between both groups was −0.18 (CI:−0.40 to −0.01) and in relative β2-AR expression between both groups was −0.3300 (CI:−0.56 to 0.00).

### 2.3. AR Gene Expression and Pfirrmann Classification

The Pfirrmann classification was used to grade IVDD preoperatively. We compared the degree of IVDD with the AR expression in the tissue of 42 IVDs. All examined discs had at least a Pfirrmann grade II. Figure 2 shows that the lowest relative AR expression was observed in IVD with low degeneration (Pfirrmann II), followed by a continuous increase to a high degree of degeneration (Pfirrmann V). A significant difference in α1A-AR (*p* = 0.005; R = 0.4274), α2A-AR (*p* = 0.026; R = 0.3547) and β2-AR (*p* = 0.005; R = 0.4376) expression was found when analyzing IVDs with low degeneration (Pfirrmann II and III; n = 15) in comparison with highly degenerated discs (Pfirrmann IV and V; n = 27). The median of differences in relative α1A-AR expression between both groups was 0.32 (CI: 0.09 to −0.74), in relative α2A-AR expression between both groups was 0.22 (CI: 0.03 to −0.51) and in relative β2-AR expression between both groups was 0.43 (CI: 0.13 to −0.71). Furthermore, relative α1A-AR (*p* = 0.038; R = 0.4016) and β2-AR (*p* = 0.025; R = 0.4297) expression was significantly increased comparing Pfirrmann grade IV with Pfirrmann grade III. The expression of α1A- (r = 0.439; *p* = 0.003), α2A- (r = 0.346; *p* = 0.023) and β2-AR (r = 0.409; *p* = 0.007) showed a positive and significant correlation with the Pfirrmann grade of IVDD.

### 2.4. AR Gene Expression and Clinical Parameters

The α1A-AR expression in disc tissue of patients with severe pain (NPRS ≥ 7.5) was significantly reduced (*p* = 0.009; R = 0.3946). The median of differences in relative α1A-AR expression between both groups was −0.39 (CI: −0.62 to −0.09). There was no significant difference in the α2A-AR expression (*p* = 0.636; R = 0.0779) and β2-AR expression (*p* = 0.061; R = 0.2835) between both groups (Figure 3a). Furthermore, medication of patients was investigated. In total, 22 patients (51.2%) received β-blockers (metoprolol, bisoprolol, nebivolol or carvedilol). The α1A-AR (*p* = 0.007; R = 0.4150) and β2-AR expression (*p* = 0.022; R = 0.3484) in disc tissue of patients who received β-blockers preoperatively were significantly reduced (Figure 3b). The median of differences in relative α1A-AR expression between both groups was −0.45 (CI: −0.79 to −0.10) and the median of differences in relative β2-AR expression between both groups was −0.30 (CI: −0.65 to −0.04). In contrast, the β2-AR expression in disc tissue of patients who received non-steroidal anti-inflammatory drugs (NSAIDs) (n = 20) preoperatively was significantly increased (*p* = 0.036; R = 0.3176) (Figure 3d). The median of differences in relative β2-AR expression between both groups was 0.37 (CI: −0.02 to 0.59). There were no significant differences in α1A-AR (*p* = 0.119; R = 0.2377) and α2A-AR expression (*p* = 0.147; R = 0.2267) between both patient cohorts. In addition, there was no significant difference in the AR expression between patients with and without nicotine abuse (Figure 3c).

No correlation between BMI and AR gene expression was found (Appendix A). In addition, there was no significant sex-dependent difference in AR gene expression (Appendix A).

### 2.5. Adjacent Segment Disease

To investigate whether ASD is associated with altered AR expression, the IVD tissues were divided into two groups (disc tissue from patients with ASD after spinal fusion surgery vs. disc tissue from patients who underwent primary spinal fusion surgery). α1A-AR expression was significantly decreased in IVD tissue from ASD (*p* = 0.041; R = 0.3111). The median of differences in relative α1A-AR expression between both groups was −0.35 (CI: −0.71 to 0.01). In contrast, no statistical difference in α2A-AR (*p* = 0.356; R = 0.1381) and β2-AR (*p* = 0.508; R = 0.1079) expression was found when analyzing IVDs from ASD in comparison with discs from primary surgery (Figure 4).

## 3. Discussion

The main causes of IVDD include biomechanical wear and tear, genetic inheritance and disturbance of physiological cellular behavior [23,24,25]. The interaction between mechanical overloading, degeneration of water-binding extracellular matrix and catabolic cell response leads to a self-reinforcing progression of degeneration. Vergroesen et al. described those interdependent processes as a “vicious circle” [2]. Despite the fact that the initiation and progress of IVDD is not fully understood, the disturbance of the interaction between mechanical behavior and cellular response seems to be an important attribute of IVDD. In this context, the SNS may play an important role.

Sympathetic nerve fibers were detected in healthy and osteoarthritic (OA) joint tissues [15,26] and Lorenz et al. showed that NE influences chondrocytes from OA cartilage regarding cell metabolism [27]. They were able to elucidate that NE induces apoptosis of chondrocytes via α1-AR and leads to decreased proliferation as well as reduced cell growth via β-AR signaling in a dose-dependent manner. The results of the study indicate that the SNS can affect cellular responses of chondrocytes. Chondrocytes from OA cartilage and from IVD are both part of the weight-bearing tissues and share several further similarities especially in the course of degeneration [20]. Following the results of Lorenz et al., the presence of sympathetic nerve fibers and the expression of ARs was detected in healthy and degenerated IVDs [28,29,30]. In addition, an enhanced and expanded expression of β2-AR in highly degenerated IVDs of mice was found [22]. To obtain more detailed information about the AR expression in IVDD, clinical and radiological parameters of the patients were collected and investigated to determine whether there is a correlation with AR expression.

An increased severity of IVDD is associated with an increase in sclerosis of the subchondral bone as well as in microscopic and macroscopic endplate damage and is highly related to LBP [31,32,33]. Consistent with these results, a significantly higher α1A-, α2A- and β2-AR expression was found in patients with inflammatory and structural changes of the vertebral endplates (MODIC II-III). Consequently, AR expression in IVDs appears to affect the vertebral endplates. This hypothesis is supported by the findings of Brown et al. who demonstrated sensory and sympathetic innervation of the vertebral endplates in patients with IVDD [34]. In order to investigate the interrelationship between AR expression during IVDD and the degeneration of vertebral endplates, future studies should also pay attention to the AR expression in cells derived from degenerated endplates.

A significant increase in α1A-, α2A- and β2-AR expression was found in highly degenerated human IVDs reflected by the above described medians of differences and CIs. The increase in AR expression might be clinically relevant with respect to sympathetic neurotransmitter-mediated catabolic processes or genesis of pain [16]. One explanation for this increase in AR expression could be their function as biomechanical sensors of cell-membrane stretch. Storch et al. described that ARs in vascular smooth muscle cells mediate vasoconstriction after stretch reception [35]. Similar mechanisms were reported in chondrocytes [36]. Accordingly, the AR upregulation might be a physiological cell response to mechanical overloading. Consequently, AR upregulation might lead to an adjustment of intradiscal pressure by cellular production of proteoglycans and their osmotic potential to bind water. Furthermore, AR-mediated cellular production of enzymes (e.g., matrix metalloproteases) could have a matrix-remodeling effect and thus adapt extracellular matrix to mechanical overloading [13,14,20,37]. Referring to the results of Lorenz et al., AR pathways can additionally reduce cell growth and initiate apoptosis depending on the concentration of neurotransmitters [27,38]. Thus, it is attractive to speculate that over-expression of ARs could lead to cell death and progression of IVDD. In this context, the knowledge of pre- and postoperative modulation of AR expression through SNS-modulating drugs could prevent progression of IVDD. This would be a promising starting point for new therapeutic strategies.

Interestingly, α1A-AR expression was significantly decreased in IVD tissue of patients with ASD after spinal fusion surgery. The interruption of sympathetic innervation of the adjacent segment IVD due to previous surgery could be an explanation for the decreased α1A-AR expression. This might cause a disturbance of cellular responses to mechanical stress and consequently aggravate degeneration. According to this hypothesis, less invasive surgery that affects the adjacent segments as little as possible could prevent the occurrence of ASD due to the reduced soft tissue damage.

In line with this assumption, α1A-AR expression was significantly reduced in IVD tissue of patients with a high preoperative pain level (NPRS ≥ 7.5) with a negative estimation of the location shift. Lumbar IVDs are thought to be an important source of LBP [39,40]. It is accepted that IVDs are segmentally innervated with sensory fibers [41,42]. Previous studies showed that a lumbar sympathetic block reduced while electrical stimulation of the lumbar sympathetic trunk provoked LBP [43,44]. Additionally, Takebayashi et al. showed that nerves in IVDs of rats do not respond to mechanical stimulation under normal conditions, but become responsive to mechanical stimuli under pathological conditions with concurrent inflammation [45]. They suggested that receptors of mechanically insensitive afferent fibers of the IVDs are silent nociceptors that modulate nociceptive information under inflammatory conditions. Due to degenerative processes and inflammation, the nociceptors become sensitive to mechanical stimuli and transmit information as discogenic LBP via the lumbar sympathetic trunk to the spinal cord. In this context, previous studies have shown that NE augments the sensitivity and activity of nociceptors in the human skin predominantly by targeting α2-AR [46,47] and that spinal application of α2-AR agonists led to analgesic effects in animal models of neuropathic pain [48,49,50]. In addition, NE contributed to a hyperalgesic joint pain in a rat arthritis model via activation of β2-AR and consequently blocking the β2-AR caused analgesia [51]. In conclusion, ARs in general play a relevant role in pain generation under inflammatory conditions. In line with this, our data suggest that α1A-AR might be important for discogenic pain. Thus, reduced AR expression might be an indicator of a reduced regenerative capacity of IVD. The disturbance of physiological cellular responses to mechanical stress might condition a progress in instability as well as perception of nociceptive information via downregulation of α1A-AR expression and consequently trigger an increase in preoperative pain level. Nevertheless, the exact relationship between pain and AR expression has not been elaborated for IVDD until now and needs to be investigated in a larger patient cohort.

Furthermore, standard medication and nicotine abuse of patients was documented and correlated to AR expression levels to investigate whether AR-modulating agents could be used as therapeutic drugs in patients having IVDD. β-blockers are mainly used for treatment of cardiovascular diseases. They can act as agonists or competitive antagonists via different intracellular signaling pathways at the same time [16,52]. Two recent studies were able to elucidate an association of hypertension with the use of β-blockers and symptomatic knee osteoarthritis [53,54]. In this context, Pasco et al. showed a bone-supporting effect of β-blocker medication in a population-based study. Patients treated with β-blockers had a lower fracture risk and a higher bone mineral density [55]. In the present study the α1A-AR and β2-AR expression was significantly reduced in disc tissue of patients who received selective or non-selective β-blockers with a negative estimation of the location shift. Comparing these results with the increasing AR expression during degeneration, it is interesting to speculate that β-blocker medication may counteract AR upregulation during degeneration and balance the physiological cellular responses to mechanical and inflammatory stress. In contrast the β2-AR expression in disc tissue of patients who received NSAIDs preoperatively was significantly increased. NSAIDs have an inhibitory effect on fracture healing as well as on other forms of postoperative bone repair via interfering prostaglandin synthesis [56,57,58]. NSAIDs also hamper osteogenic cell proliferation by inhibiting angiogenesis [59]. The results of this study may indicate that NSAIDs increase AR expression during degeneration and thus condition a progress in degeneration. Furthermore, nicotine is known to stimulate sympathetic neurotransmission via a norepinephrine-releasing effect which induces β- and α-AR signaling pathways [60]. Since 26% of the patients in this study consume nicotine, the influence of nicotine abuse on AR expression was analyzed. There was no significant difference in α- and β-AR expression between patients with and without nicotine abuse. This may indicate that nicotine plays a minor role in the pathogenesis of IVDD. Nevertheless, the present study design cannot give a final answer to this question and the effect of nicotine on AR expression in IVDD remains unclear.

In summary, SNS-modulating drugs might have an impact on AR expression in IVD tissue. Although the exact relationship needs to be further investigated, detailed knowledge about SNS-modulating drugs that further aggravate IVDD would improve patient outcomes. In this context, the dose-dependent effects at the cellular level as well as the cellular mechanisms are of particular interest to estimate the use of AR-modulating drugs as therapeutic options. Possible adverse effects of systematically applied AR antagonists and agonists have to be taken into account as well. To reduce the risks of adverse effects, local application of AR modulating drugs should be an additional task of future research.

There are a few limitations of this study. One limitation is the absence of a control group. For ethical reasons, it is not justifiable to violate an undegenerated human disc by taking a sample for study reasons. Since no healthy human IVD tissues (Pfirmann grade 0) were available, it was not possible to compare healthy with degenerated stages. Nevertheless, a similar comparison was performed by Kupka et al. in a murine model and an enhanced and locally expanded AR expression in highly degenerated IVDs of mice was found [22]. In addition, the present study demonstrated a significant positive correlation between the degree of IVDD and AR expression, suggesting that healthy discs do not show an increased AR expression. Another limitation could have been the relatively high mean BMI (30.77 kg/m^2^) of our patient cohort. Under the assumption that a high BMI is associated with an increased load on the IVD, this might have an influence on the AR gene expression. Nevertheless, we found no correlation between BMI and AR gene expression in our patient cohort (Appendix A). However, we cannot exclude that much lower BMI levels than in our cohort could influence AR gene expression. Moreover, there is an imbalance between the number of male (n = 9) and female (n = 34) patients in our cohort. Also in this context, we found no significant sex-dependent differences in AR gene expression (Appendix A). This study was not specifically designed to evaluate the influence of BMI and gender on the AR gene expression and those hypotheses should be investigated in further studies. Lastly, the underlying causes of IVDD were heterogeneous and may affect AR expression in IVD tissue in varying ways. This study was not specifically designed to evaluate the influence of the underlying diseases on the AR expression and those hypotheses should be investigated in a larger study population. Nonetheless, all patients suffered from a degenerated IVD and the comparison between AR expression in IVDs from primary and revision surgery allowed a more detailed investigation of subgroups. In this context the ASD subgroup is particularly interesting because ASD is still an unsolved problem in spinal fusion surgery.

In conclusion, the results of this study indicate that a relationship between IVDD and AR expression exists. Thus, the SNS and its neurotransmitters might play a role in IVDD pathogenesis. A dysregulation of AR expression might lead to a disturbance of homeostasis in the IVD and thus aggravate intra- and extracellular processes of degeneration. In future studies, the functional response of isolated IVDs to sympathetic neurotransmitters needs to be investigated. The results will contribute to a better understanding of cellular processes of IVDD and novel therapeutic approaches.

## 4. Materials and Methods

### 4.1. Human IVD Tissue Samples and Patient Characteristics

Lumbar IVD samples were collected during disc preparation for an interbody cage from 43 patients undergoing spinal fusion surgery. The samples were anonymized and stored at −80 °C until analysis. Patient characteristics as well as the preoperative radiological classifications of IVDD were obtained retrospectively.

### 4.2. AR Gene Expression

The gene expression of all AR subtypes (α1A, α1B, α1D, α2A, α2B, α2C, β1, β2 and β3) and tyrosine hydroxylase was measured by semi-quantitative reverse transcriptase polymerase chain reaction (RT-PCR). The intraoperatively obtained IVD tissue was minced with a scalpel and further processed according to manufacturer’s instructions (NucleoSpin^®^ RNA/Protein, Macherey and Nagel, Düren, North Rhine-Westphalia, Germany). Reverse transcription of mRNA was performed with Quantabio qScript cDNA Synthesis Kit (VWR, Darmstadt, Hessen, Germany) and 5 ng of cDNA per optimized primer was used for PCR (Taq PCR Master Mix, Qiagen, Hilden, North Rhine-Westphalia, Germany). Validated primer sets obtained from Thermo Fisher Scientific (Darmstadt, Hessen, Germany) were used to amplify human ARs (Table 2). The results were visualized by agarose gel electrophoresis with GelRed (Biotium, Fremont, CA, USA) and ChemiDoc XRS+ (BioRad, Dreieich, Hessen, Germany) (Appendix A). We used human GAPDH as a housekeeping gene for normalization. The RT-PCR gels were analyzed densitometrically based on the intensities of corresponding PCR bands using the Image Lab software (BioRad, Dreieich, Hessen, Germany). GAPDH band intensity of each individual patient was defined as “1” and AR expression levels were calculated in relation to that value. In the following, only the α1A-, α2A- and β2-ARs are considered, since their analysis with clinical parameters yielded significant results.

### 4.3. Radiological Classification of IVDD and Patient Characteristics

To evaluate degenerative and inflammatory changes of the vertebral endplates and adjacent vertebral bodies, preoperative Modic classification of the respective segment was determined by MRI [61,62]. The Pfirrmann classification was used to grade IVDD preoperatively on MRI T2 spin-echo sequence weighted images (Table 3) [63]. In addition, patient characteristics for a total n = 43 patients were collected retrospectively. The pre- and postoperative back pain level of patients was determined with the NPRS which ranges from ‘0’ representing “no pain” to ‘10’ representing “the worst pain imaginable”. To distinguish between patients with severe back pain and low or moderate back pain, patients were divided into two groups, following the cut-off points of Boonstra et al. [64]: Group 1: NPRS < 7.5 (low or moderate pain) and Group 2: NPRS ≥7.5 (severe pain). Pre-existing conditions and levels of previous surgeries were recorded. Furthermore, preoperative medication and nicotine abuse of patients were collected to investigate whether drugs that affect the SNS are associated with changes in AR gene expression (Appendix A).

### 4.4. Ethical Approval

The Ethics Committee of the University Hospital Frankfurt (Goethe University Frankfurt, Germany) has approved the study under the number 19/525 and confirmed that no informed consent is required due to the retrospective design. All experiments were performed in accordance with relevant guidelines and regulations. The data were anonymized before use.

### 4.5. Statistical Analysis

The Shapiro-Wilk test was used to test normal distribution of the analyzed parameters. Nonparametric independent variables were compared with Mann–Whitney tests. Effect size (R) was calculated according to Rosenthal [65]. The Hodges–Lehmann method was used to estimate median differences with 95% confidence intervals (CI) for the respective parameters [66]. A non-parametric Levene’s test was used to verify the equality of variances in the samples (homogeneity of variances). A Spearman’s rank correlation coefficient (r) was calculated to identify relationships between AR expression and IVDD. Statistical data analysis was performed with SPSS version 26 (IBM Corporation, New York, NY, USA) and BiAS. für Windows version 11.12 (H. Ackermann, epsilon-Verlag, Darmstadt-Hochheim, Germany). The significance level was set at *p* ≤ 0.05.

## Figures and Tables

**Figure 1 ijms-23-15358-f001:**
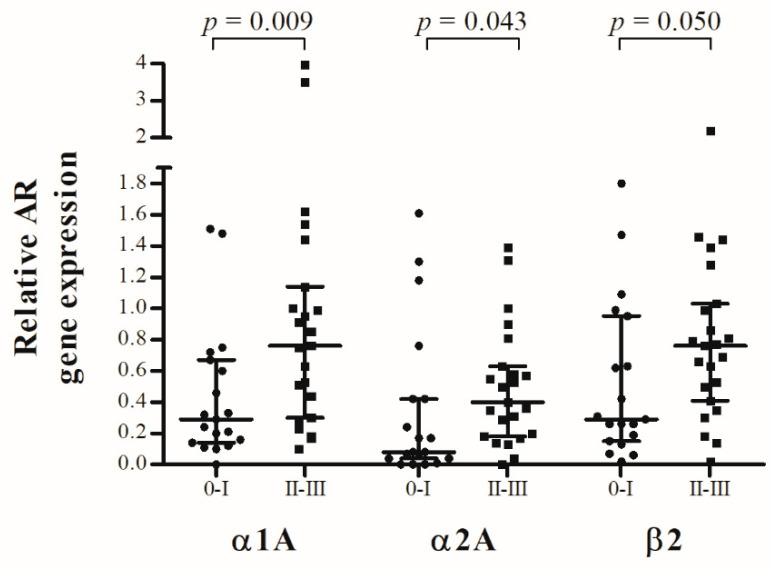
Correlation between relative AR gene expression in IVD tissue and radiological changes of the vertebral endplates (Modic classification). Modic 0 and I (black circle) as well as Modic II and III (black square) were each assigned to one group. Each black reference point represents an individual patient (n = 42). Data represent medians with interquartile range.

**Figure 2 ijms-23-15358-f002:**
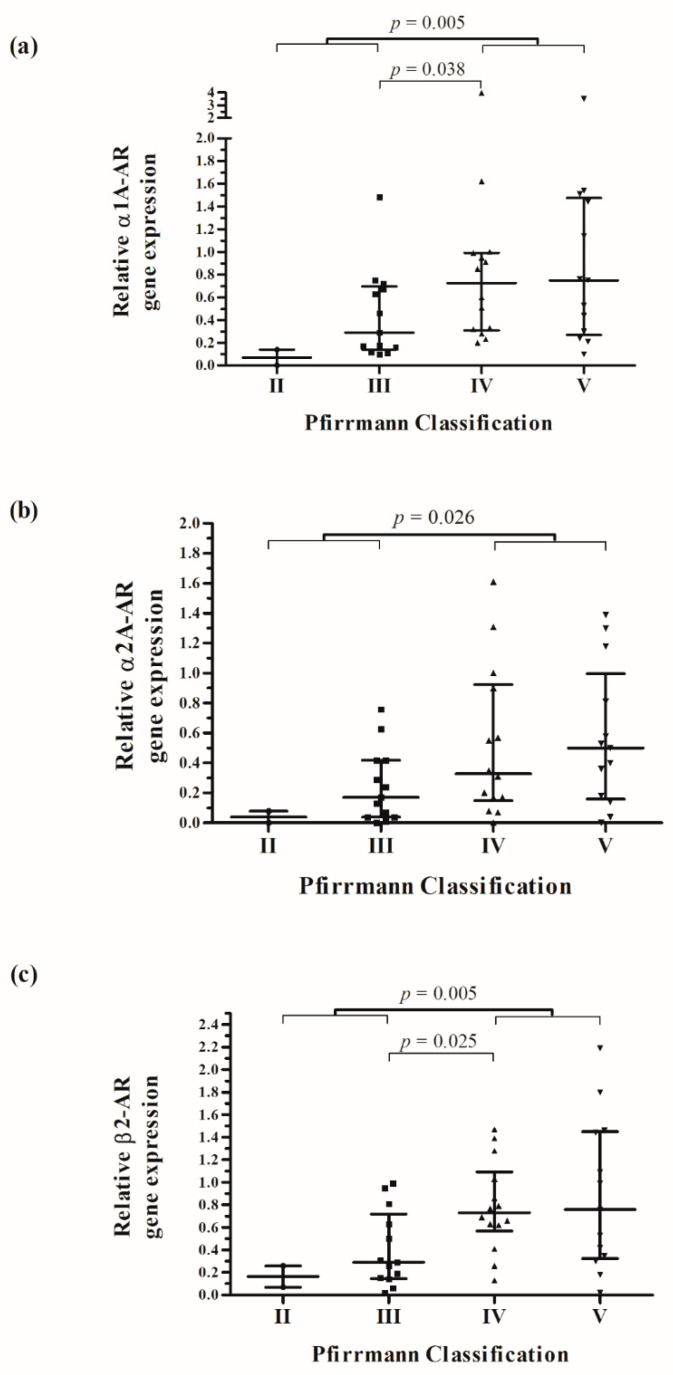
Correlation between relative α1A- (**a**), α2A- (**b**) and β2-AR gene expression (**c**) in IVD tissue and radiological grade of IVDD (Pfirrmann classification). Each black reference point represents an individual patient (n = 42). Data represent medians with interquartile range.

**Figure 3 ijms-23-15358-f003:**
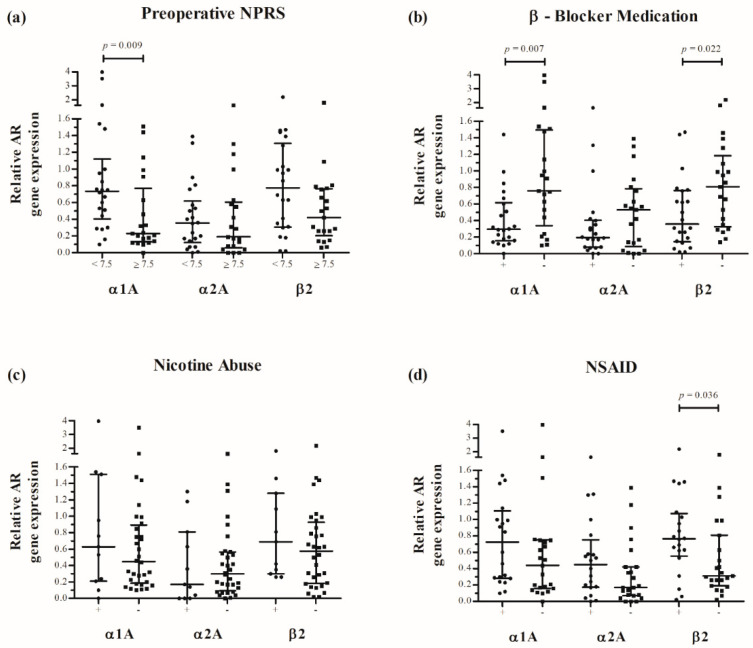
Correlation between relative AR gene expression in IVD tissue and preoperative pain level (**a**). Patients were divided by Numeric Pain Rating Scale (NPRS) responses of low or moderate pain (<7.5—black circle) and severe pain (≥7.5—black square). Correlation between relative AR gene expression in IVD tissue and β-blocker medication (**b**, **“+”**), nicotine abuse (**c**, **“+”**) as well as NSAID medication (**d**, **“+”**). Each black reference point represents an individual patient (n = 43). Data represent medians with interquartile range.

**Figure 4 ijms-23-15358-f004:**
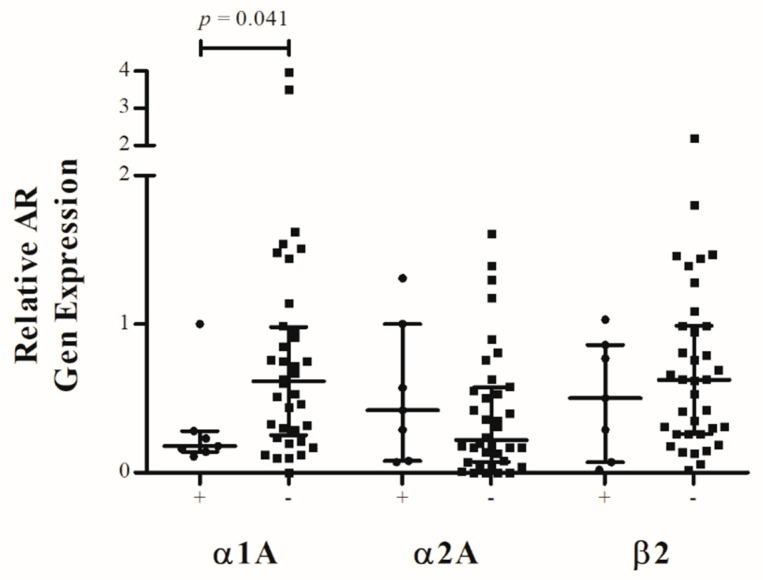
Correlation between relative AR gene expression in IVD tissue of patients with (“+”, black circle) and without (“−”, black square) Adjacent Segment Disease. Each black reference point represents an individual patient (n = 43). Data represent medians with interquartile range.

**Table 1 ijms-23-15358-t001:** Patient Characteristics, Radiological Classification Systems and Medication.

Patient Characteristics	
Patients, n	43
Sex, no. female (%)/no. male (%)	34 (79.1%)/9 (20.9%)
Age [years], mean (SD)	69.23 (7.66)
BMI [kg/m^2^], mean (SD)	30.77 (5.67)
Preoperative NPRS, mean (SD)	6.81 (1.53)
Postoperative NPRS, mean (SD)	2.40 (1.61)
Adjacent segment disease, n (%)	7 (16.3%)
Nicotine abuse, n (%)	11 (25.6%)
Medication: NSAID, n (%)	20 (46.5%)
Medication: β-blocker, n (%)	22 (51.2%)
Level of operation:	
L 1/2	2 (4.7%)
L 2/3	6 (14.0%)
L 3/4	9 (20.9%)
L 4/5	15 (34.9%)
L5/S1	11 (25.6%)
Total	43 (100%)
Modic classification, n (%)	
0	10 (23.3%)
I	9 (20.9%)
II	20 (46.5%)
III	3 (7.0%)
Missing MRI	1 (2.3%)
Total	43 (100%)
Pfirrmann classification, n (%)	
I	0 (0%)
II	2 (4.7%)
III	13 (30.2%)
IV	14 (32.6%)
V	13 (30.2%)
Missing MRI	1 (2.3%)
Total	43 (100%)

**Table 2 ijms-23-15358-t002:** The primers used for PCR.

Gene Symbol	NCBI Reference	Forward (5′-3′)	Reverse (5′-3′)
GAPDH	NM_001289745.2	CTCCTGTTCGACAGTCAGCC	TTCCCGTTCTCAGCCTTGAC
ADRA1A	NM_000680.3	CCATGCTCCAGCCAAGAGTT	TCCTGTCCTAGACTTCCTCCC
ADRA1B	NM_000679.3	GTCCACCGTCATCTCCATCG	GAACAAGGAGCCAAGCGGTAG
ADRA1D	NM_000678.3	TGACTTTCCGCGATCTCCTG	TTACCTGCCACGGCCATAAG
ADRA2A	NM_000681.3	TGGTCATCGGAGTGTTCGTG	GCCCACTAGGAAGATGGCTC
ADRA2B	NM_000682.6	GACATTTCACCGGCAACACC	GGGACTGAGAACCAGGAAGC
ADRA2C	NM000683.3	CGATGTGCTGTTTTGCACCT	GGATGTACCAGGTCTCGTCG
ADRB1	NM_000684.2	TAGCAGGTGAACTCGAAGCC	ATCTTCCACTCCGGTCCTCT
ADRB2	NM_000024.5	CAGAGCCTGCTGACCAAGAA	GCCTAACGTCTTGAGGGCTT
ADRB3	NM_000025.2	GCCAATTCTGCCTTCAACCC	GCCAGAGGTTTTCCACAGGT
TH	NM_000360.3	CAGGCAGAGGCCATCATGT	GTGGTCCAAGTCCAGGTCAG

**Table 3 ijms-23-15358-t003:** Modic classification of endplate changes and Pfirrmann classification of lumbar intervertebral disc degeneration.

Modic Classification				
Vertebral Endplates	Histopathology	Signal intensity		
Modic I	Marrow edema	T1 hypointense, T2 hyperintense		
Modic II	Fatty degeneration	T1 hyperintense, T2 iso-/hyperintense		
Modic III	Subchondral bony sclerosis	T1 hypointense, T2 hypointense		
**Pfirrmann classification**				
Grade	Structure	Signal intensity (T2)	Distinction of Nucleus and Anulus	Height
Pfirrmann I	homogeneous, white	hyperintense	clear	normal
Pfirrmann II	inhomogeneous, horizontal bands	hyperintense	clear	normal
Pfirrmann III	inhomogeneous, gray	intermediate	unclear	normal– slightly decreased
Pfirrmann IV	inhomogeneous, gray to black	intermediate to hypointense	lost	slightly– moderately decreased
Pfirrmann V	inhomogeneous, black	hypointense	lost	collapsed

## Data Availability

The datasets generated during and/or analyzed during the current study are available from the corresponding author on reasonable request.

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
