# Peer review of "Correlation between Adrenoceptor Expression and Clinical Parameters in Degenerated Lumbar Intervertebral Discs"

_ijms, 2022, doi:10.3390/ijms232315358_

Round 1
Reviewer 1 Report
Dear authors,
although i do appreciate the importance and the logic of your study as well as the efforts you put into it, the manuscript has some serious shortcomings that require major revision. Unfortunately the presentation of the results (Figures) is inadequate, as it is very hard to determine the values of the medians and the IQRs due to the coarseness of the y-axis scales. You could offer a table with the values as you did for the descriptive statistics.
Also, you quantify many DNA gels, but you don't provide images that one could appraise, which is really important for semi quantitative methods. Please provide images of the gels as supplemental material.
The document reads quite well, however, in the discussion you seem to use "furthermore" quite frequently which doesn't make for good style.
Some questions that occured to me while reading your manuscript:
How did you decide on the grouping of low-moderate vs severe pain? Why not have three groups?
You have a female/male imbalance and 85% of the subjects have a BMI >25, is that relevant for the results? Please discuss
In line 84 you state: most of the IVD samples were collected from L4/5 but then they say they account for 34.9%. That is not even half of the samples. Maybe you meant "the largest proportion of the samples"?
Best Regards
Author Response
We thank the Reviewers for their valuable comments. We set a high value on meeting the requirements of the Reviewers, especially regarding the following points:
Reviewer 1: Unfortunately the presentation of the results (Figures) is inadequate, as it is very hard to determine the values of the medians and the IQRs due to the coarseness of the y-axis scales. You could offer a table with the values as you did for the descriptive statistics.
Response: Thank you for this comment. We have changed the scale of the respective y-axes. Furthermore, we have provided a new table as supplemental material, which includes the respective medians and IQRs (Table S1).
Reviewer 1: Also, you quantify many DNA gels, but you don't provide images that one could appraise, which is really important for semi quantitative methods. Please provide images of the gels as supplemental material.
Response: Images of representative gels are now provided as supplemental material (Figure S1).
Reviewer 1: The document reads quite well, however, in the discussion you seem to use "furthermore" quite frequently which doesn't make for good style.
Response: Thank you for this comment. We have revised the wording to improve the style of our manuscript.
Reviewer 1: Some questions that occured to me while reading your manuscript:
How did you decide on the grouping of low-moderate vs severe pain? Why not have three groups?
Response: Thank you for this question. Our aim was to investigate, whether severe pain is associated with altered AR expression. In our patient cohort only 8 patients had a preoperative pain ≤ 5 and only one patient had a preoperative pain ≤ 3. In order to prevent the analysis of too small group sizes and to investigate similar group sizes, we decided to divide patients in groups with severe back pain (n=21) and low or moderate back pain (n=22) following the cut-off points of Boonstra et al. [64].
We described this approach in the methods section (Line 378-384).
Reviewer 1: You have a female/male imbalance and 85% of the subjects have a BMI >25, is that relevant for the results? Please discuss
Response: Thank you for this comment. In order to investigate the influence of BMI and gender on AR expression we have performed additional statistical analyses. No correlation between BMI and AR gene expression was found (Figure S2). In addition, there was no significant difference in AR gene expression between males and females (Figure S3). Nevertheless, we included both points into the limitations section (Results: Line 162-164 AND Discussion: 321-330).
Reviewer 1: In line 84 you state: most of the IVD samples were collected from L4/5 but then they say they account for 34.9%. That is not even half of the samples. Maybe you meant "the largest proportion of the samples"?
Response: Thank you for this comment. Yes, we meant “the largest proportion of the IVD samples”. We have changed the wording to: “The largest proportion of the IVD samples was collected from the segment L4/5 (34.9%).” (Line 89-90)
Sincerely,
On behalf of the authors
Reviewer 2 Report
Reviewer’s comments
Manuscript ID: ijms-1922086
Title: Correlation Between Adrenoceptor Expression and Clinical Parameters in Degenerated Lumbar Intervertebral Discs.
In the current manuscript, authors investigated the relationship of expression level of the detected Ars in IVDs with clinical parameters of the patients such as radiological degree of degeneration, preoperative pain level and medication and suggested that a relationship between IVDD and AR expression exists.
This is an interesting study and the paper is generally well written and structured.
The paper is very similar to Reference 22 [Kupka, J.; Kohler, A.; El Bagdadi, K.; Bostelmann, R.; Brenneis, M.; Fleege, C.; Chan, D.; Zaucke, F.; Meurer, A.; Rickert, M.; et al. Adrenoceptor Expression during Intervertebral Disc Degeneration. Int. J. Mol. Sci. 2020, 21, doi:10.3390/ijms21062085.]. Some authors’ names are in both papers. The number of included patients is same (n=43) and the experimental design is similar (relationship between IVDD and AR expression)
Minor points.
Line 82: NPRS needs to be written in full name.
Line 312: IRB number is mandatory.
Author Response
We thank the Reviewers for their valuable comments. We set a high value on meeting the requirements of the Reviewers, especially regarding the following points:
Reviewer 2: The paper is very similar to Reference 22 [Kupka, J.; Kohler, A.; El Bagdadi, K.; Bostelmann, R.; Brenneis, M.; Fleege, C.; Chan, D.; Zaucke, F.; Meurer, A.; Rickert, M.; et al. Adrenoceptor Expression during Intervertebral Disc Degeneration. Int. J. Mol. Sci. 2020, 21, doi:10.3390/ijms21062085.]. Some authors’ names are in both papers. The number of included patients is same (n=43) and the experimental design is similar (relationship between IVDD and AR expression)
Response: Thank you for this question. The first description of AR expression in the IVD has been published recently by us in IJMS: “Kupka J, Kohler A, El Bagdadi K, Bostelmann R, Brenneis M, Fleege C, Chan D, Zaucke F, Meurer A, Rickert M, Jenei-Lanzl Z. Adrenoceptor Expression during Intervertebral Disc Degeneration. International journal of molecular sciences. 2020;21.”
Because of the interesting and promising results, we now performed an in-depth analysis of the patients' clinical parameters (Modic classification, Pfirrmann classification, medication, pre-existing diseases, numeric pain rating scale) and aimed to correlate the AR expression data with these parameters. To the best of our knowledge, this is the first study investigating the AR receptor profile in human IVDs to test the interrelationship between specific receptor subtypes and clinical parameters of the patients. The results of this study indicate that a relationship between IVD degeneration and AR expression exists and that the SNS and its neurotransmitters might play a role in IVD degeneration and cellular responses. We believe that these new findings contribute to a better understanding of cellular processes and the development of new drugs targeting the autonomic nervous system for IVDD treatment.
Minor points.
Reviewer 2: Line 82: NPRS needs to be written in full name.
Response: Thank you for this comment. We included the full name (Line 87).
Reviewer 2: Line 312: IRB number is mandatory.
Response: Thank you. The IRB number is mentioned under 4.4 Ethical approval (Line 391):
“The Ethics Committee of the University Hospital Frankfurt (Goethe University Frankfurt, Germany) has approved the study under the number 19/525 and confirmed that no informed consent is required due to the retrospective design.”
Additional changes:
During the adjustments of the figures we recognized that four patients were assigned to the wrong Modic group. After correction, we found a significant difference between the respective Modic groups. These findings support our hypothesis that AR expression is associated with disc and cartilage degeneration. We have incorporated the corrections in the results (Line 100-112) and discussion section (Line 211-222).
Sincerely,
On behalf of the authors
Reviewer 3 Report
This is a very interesting follow-up study using samples from a previous publication (Kupka et al. Int. J. Mol. Sci. 2020, 21): Adrenoreceptor expression during intervertebral disc degeneration.
There is some overlapping content with this paper - but the present study is considering now a possible correlation of the expression levels detected in the previous study in tissue of degenerated IVDs with clinical parameters of the patients.
A relationship between disc degeneration and adrenoreceptor expression could be found. The authors conclude that the sympathetic nervous system and its neurotransmitters might play a role in IVDD pathogenesis.
The manuscript is well-written and already very close to being suitable for publication. Some minor changes can be recommended to further improve it:
Methods section, 2.2 AR gene expression and Modic classification:
The gene expression levels are given with four digits - two digits are sufficient.
The absence of a control group is indeed a limitation of this study - but the authors well explain the reasons. It is important that the authors mention in their limitations also the high BMI of most patients and the majority of patients being female. But, as the study was not designed for investigation of BMI and gender influences, this can be accepted.
Author Response
We thank Reviewer 3 for his/her valuable comments. Based on his/her recommendations as well as on his/her constructive critique, we were able to further improve our manuscript. We set a high value on meeting the requirements of Reviewer 3:
Reviewer 3: Methods section, 2.2 AR gene expression and Modic classification: The gene expression levels are given with four digits - two digits are sufficient.
Response: Thank you for this comment. We have changed the digits according to your suggestion.
Reviewer 3: The absence of a control group is indeed a limitation of this study - but the authors well explain the reasons. It is important that the authors mention in their limitations also the high BMI of most patients and the majority of patients being female. But, as the study was not designed for investigation of BMI and gender influences, this can be accepted.
Response: Thank you for this comment. In order to investigate the influence of BMI and gender on AR expression we have performed additional statistical analyses. No correlation between BMI and AR gene expression was found (Figure S2). In addition, there was no significant difference in AR gene expression between males and females (Figure S3). Nevertheless, we included both points into the results section (Line 161-164) and the limitations section (BMI Line 318-323 AND female/male imbalance Line 323-327).
Round 2
Reviewer 1 Report
Dear authors,
Thank you for the resubmission of the revised manuscript. I appreciate the additional results and supplemental data. I was, however, puzzled on why the results in chapter 2.2 on modic changes are now significant and weren't before. What was the issue?
best regards
Author Response
Thank you for your valuable feedback.
During the adjustments of the figures we recognized that four patients were assigned to the wrong Modic group. After correction, we found a significant difference between the respective Modic groups. These findings support our hypothesis that AR expression is associated with disc and cartilage degeneration. We have incorporated the corrections in the results (Line 100-112) and discussion section (Line 211-222).
Reviewer 2 Report
Although the authors stated that they investigated the expression of adrenal receptors according to different disc degeneration, previous their study seemed to already investigate the expression of adrenal receptors according to different disc degeneration.
Author Response
Thank you for your Feedback.
As commented by reviewer 3, this is a interesting follow-up study using samples from a previous publication. There is some overlapping content with this paper - but the present study is considering now a possible correlation of the expression levels detected in the previous study in tissue of degenerated IVDs with clinical parameters of the patients.